# Monitoring the combined effects of drought and salinity stress on crops using remote sensing in the Netherlands

Wen Wen [1, *], Joris Timmermans[1, 2, 3], Qi Chen[1] and Peter M. van Bodegom[1]

[1]Institute of Environmental Sciences (CML), Leiden University, Box 9518, 2300 RA Leiden, The Netherlands
[2]Institute for Biodiversity and Ecosystem Dynamics, University of Amsterdam, 1090 GE Amsterdam, The Netherlands
[3]Lifewatch ERIC, vLab & Innovation Centre, 1090 GE Amsterdam, The Netherlands

*Correspondence:* Wen Wen (w.wen@cml.leidenuniv.nl)

**Abstract.** Global sustainable agricultural systems are under threat, due to increasing and co-occurring drought and salinity stresses. Combined effects of these stresses on agricultural crops have traditionally been evaluated in small-scale experimental studies. Consequently, large-scale studies need to be performed to increase our understanding and assessment of the combined impacts in agricultural practice in real-life scenarios. This study aims to provide a new monitoring approach using remote sensing observations to evaluate the joint impacts of drought and salinity on crop traits. In our tests over the Netherlands at large spatial (138.74 km$^2$), we calculated five functional traits for both maize and potato from Sentinel-2 observations, namely: leaf area index (LAI), the fraction of absorbed photosynthetically active radiation (FAPAR), the fraction of vegetation cover (FVC), leaf chlorophyll content (Cab) and leaf water content (Cw). Individual and combined effects of the stresses on the seasonal dynamics in crop traits were determined using both one-way and two-way ANOVAs. We found that both stresses (individual and co-occurring) affected the functional traits of both crops significantly (with R$^2$ ranging from 0.326 to 0.796), though with stronger sensitivities to drought than to salinity. While we found exacerbating effects within co-occurrent stresses, the impact-level depended strongly on the moment in the growing season. For both crops, LAI, FAPAR and FVC dropped the most under severe drought stress conditions. The patterns for Cab and Cw were more inhibited by co-occurring drought and salinity. Consequently, our study constitutes a way towards evaluating drought and salinity impacts in agriculture with the possibility of potential large-scale application for sustainable food security.

**Keywords:** Drought; Salinity; Agriculture; Remote sensing; Functional traits

## 1 Introduction

Food production is required to increase by 70% to satisfy the growing population demand by the year 2050 (Godfray et al., 2010). Meanwhile, food security is becoming increasingly threatened due to the increasing abiotic stresses under the influence of global climate change; abiotic stresses including drought, soil salinity, nutrient stress and heavy metals are estimated to constrain crop productivity by 50% ~ 80% (Shinozaki et al., 2015). Of these stresses, drought and salinity stress have been identified as the two main factors to limit crop growth, affecting respectively 40% and 11% of the global irrigated areas (Dunn et al., 2020; FAO, 2020). With drought and salinity forecasted to increase spatially and in severity (Rozema and Flowers, 2008; Schwalm et al., 2017; Trenberth et al., 2013), and with predictions of higher co-occurrence around the world (Corwin, 2020; Jones and van Vliet, 2018; Wang et al., 2013), food production will be more deeply challenged by both stresses.

Numerous small-scale experimental studies for a large variety of crops have shown that the impact of co-occurring drought and salinity stress is exacerbated. Co-occurrence of drought and salinity stress is found to decrease the yield of spinach (Ors and Suarez, 2017) and the forage grass *Panicum antidotale* (Hussain et al., 2020) more compared with the occurrence

of one of these stresses only. Likewise, cotton root growth tends to be more inhibited under the co-occurrence of drought and salinity than by isolated occurrences (Zhang et al., 2013). Similarly, the exacerbating effect of co-occurring stresses limits both maize reproductive growth and grain formation (Liao et al., 2022). While these studies demonstrate the exacerbating effects of co-occurring drought and salinity stress, they have limitations in projecting the impact towards real farmers' conditions due to their small-scale experimental nature. Thus, there is still a significant knowledge gap concerning the large scale evaluation of the combined impacts of drought and salinity.

Remote sensing (RS) provides a huge potential to close this knowledge gap due to its capability to monitor continuous large areas at a frequent interval. For this, remote sensing has traditionally used vegetation indices, such as Normalized Difference Vegetation Index (NDVI) (Tucker, 1979). However, such indices provide limited information on how the impact is achieved (e.g. Wen et al., 2020) and how it can be mitigated. With the launch of better multispectral and high-resolution satellite sensors (such as Sentinel-2), new RS methods (e.g., hyperspectral, thermal infrared, microwave) have been identified to detect stress in both natural vegetation (Gerhards et al., 2019; Vereecken et al., 2012) as well as for agricultural applications (Homolova et al., 2013; Weiss et al., 2020). Specifically, these new RS methods allow for the retrieval of plant traits that directly link to plant processes, such as leaf biochemistry and photosynthetic processes, and thereby provide high potential for agricultural applications. RS plant traits of specific interest to monitor crop health include leaf area index (LAI) (Wengert et al., 2021), canopy chlorophyll content (Cab*LAI) (Gitelson et al., 2005), canopy water content (Cw*LAI) (Kriston-Vizi et al., 2008), the fraction of absorbed photosynthetically active radiation (FAPAR) (Zhang et al., 2015) and the fraction of vegetation cover (FVC) (Yang et al., 2018). Canopy chlorophyll content and mean leaf equivalent water thickness (EWT) of maize differed remarkably under drought stress using hyperspectral remote sensing data (Zhang and Zhou, 2015). Using a look-up-table approach, LAI and chlorophyll content of wheat obtained from a radiative transfer model showed potential to assess drought levels (Richter et al., 2008). However, while there have been several attempts to monitor the response of crop health with either a drought or salinity focus, not much research has taken these factors into account simultaneously (Wen et al., 2020).

In this study, we propose a novel approach to estimate, compare and evaluate the impacts of drought, salinity, and their combination on crop traits using remote sensing. To allow for a detailed evaluation of this approach we applied it to analyze the impacts of the 2018 summer drought in the Netherlands on agricultural crops. In this, a stress co-occurrence map was created by overlaying a high-resolution drought map of 2018 with a groundwater salinity map. Then, we characterized the response of maize and potato to different stress conditions based on five plant traits (LAI, FAPAR, FVC, Cab, and Cw). Two-way ANOVAs were adopted to test the main effects and the interactive effect between stress combinations and time on crop traits. Moreover, the effect of drought and salinity on crop traits was determined across the growing season with one-way ANOVAs. Consequently, this approach facilitates simultaneously monitoring crop health at various scales (regional, national, continental) across multiple stresses (drought, salinity) and multiple species.

## 2 Methodology

To achieve our aim of monitoring the impacts of (co-occurring) drought and salinity on agricultural production, we developed a new approach to estimate crop traits from remote sensing observations. Specifically, we developed an approach that integrates image-processing techniques, such as image classification, co-registration, land surface parameter retrieval, and time-series analysis. Using these techniques, we were able to estimate the drought, salinity, and crop growth.

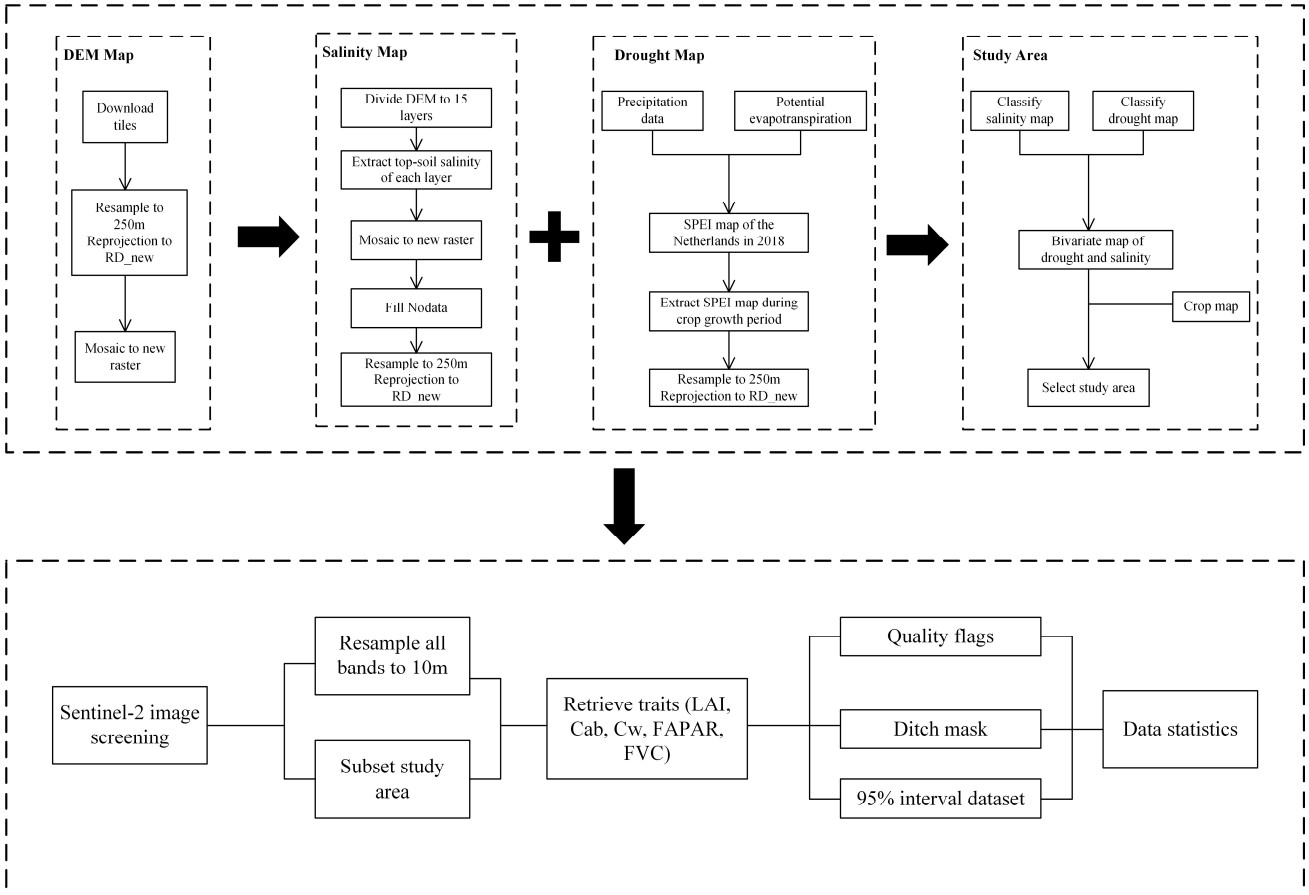


**Figure 1.** Technical workflow of the maps and data framework.
To allow for a detailed evaluation, we focused on the 2018 summer drought in the Netherlands. This period was selected
because of the extreme drought that affected a large part of Europe (Masante D., 2018). Within parts of the selected area
salinity was reported to increase during that same period (Broekhuizen, 2018). Hence this study area provides us with the
opportunity to investigate the combined impacts of these stresses on crops. In the following paragraphs, we provide more
information on the specific processing steps.
**2.1 Study area and data**
**2.1.1 Drought map**
A drought map of the Netherlands in 2018 was created based on the standardized precipitation evapotranspiration index
(SPEI) drought index, which was calculated from long-term precipitation data and potential evapotranspiration, from 2004
to 2018 (Chen et al., 2022). Specifically, SPEI was estimated using a 3-month sliding time window, as this was found best
to investigate the impacts on the local ecosystems. We have extracted SPEI-3 data from April 1st to October 30th, totally
214 days, as this coincided with the crop growth period of both maize and potato. Then, the drought map was resampled
to 250m resolution using the nearest neighbor interpolation and reprojected to RD_new projection. The RD_new projection
(EPSG:28992) is a projected coordinate reference system of the Netherlands. All maps were projected to RD_new
projection to create consistent data layers. We defined -1 and -1.5 as daily thresholds for different drought severity classes
according to previous classifications (McKee et al., 1993; Tao et al., 2014). Thus, (cumulative) SPEI for no drought should
be between -214 to 0, SPEI for moderate drought should be between -321 to -214 and for severe drought, SPEI should be
lower than -321 when calculated for the whole growing period (Fig. 2a).

### 2.1.1 Salinity map

A top-soil salinity map of the Netherlands was created based on a nationwide fresh-salt groundwater dataset, which derived chloride concentrations as a salinity indicator (https://data.nhi.nu/). To obtain the salinity map of the top-soil, 15 layers of the groundwater salinity were extracted from the 3D groundwater salinity map. For each location, the layer closest to the corresponding to location's elevation (according to the Digital Elevation Model), i.e. closest to the soil surface, was selected. The salinity map was resampled to 250 m resolution and reprojected to RD_new projection. Ultimately, the salinity map was classified into three levels namely no-salinity (0.1 $g·L^{-1}$ to 0.8 $g·L^{-1}$), moderate salinity (0.8 $g·L^{-1}$ to 2.5 $g·L^{-1}$), severe salinity ($>= 2.5$ $g·L^{-1}$) according to the salt-resistant capacity of various crops cultivated in the Netherlands (Mulder et al., 2018; Stuyt, 2016) (Fig. 2b).

### 2.1.3 Crop map

The crop map of the Netherlands in 2018 was collected from the Key Register of Parcels (BRP) of the Netherlands Enterprise Agency (https://www.pdok.nl/introductie/-/article/basisregistratie-gewaspercelen-brp-). The crop map was resampled to 250m resolution and reprojected to RD_new projection (Fig. 2d).

### 2.1.4 Co-occurrence map of drought and salinity

The drought map and the salinity map were overlain to evaluate co-occurrences of drought and salinity of the Netherlands in 2018 (Fig. 2c). By classifying the three stress levels for the individual occurrences, we obtained nine stress classes of co-occurring drought and salinity, namely no stress, moderate drought only (MD), severe drought only (SD), moderate salinity only (MS), severe salinity only (SS), moderate drought and moderate salinity (MD+MS), moderate drought and severe salinity (MD+SS), severe drought and moderate salinity (SD+MS), and severe drought and severe salinity (SD+SS).

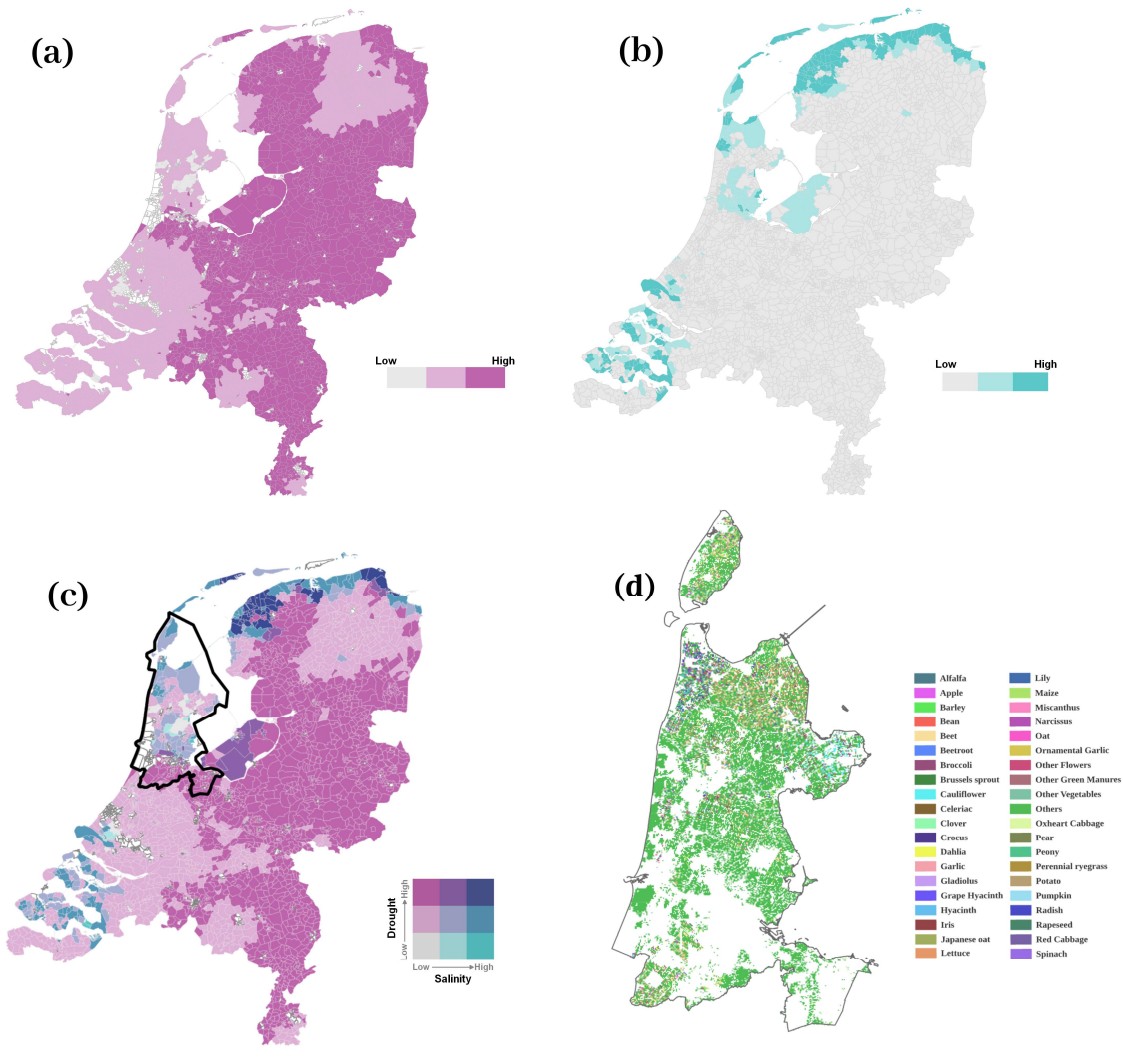

**Figure 2.** Map of the Netherlands overlaying a) drought and b) salinity to show c) the co-occurrence of drought and salinity in 2018. The selected study area is indicated by black lines in panel c. d) The associated crop map of the study area in 2018.

### 2.1.5 Study area selection

Based on the national map of the Netherlands (Fig. 2c), a single region with similar soil type, climate, tillage systems, and irrigation methods was chosen to minimize the interference of these factors on the observed trait expressions. The province of North-Holland was selected because it contained the most (7 out of 9) combinations of drought and salt stress (Fig. 2c), namely: no stress, MD, SD, MS, SS, MD+MS, and SD+SS. Moreover, both maize and potato were cultivated across all stress combinations in this province. For further analysis, MS and SS were grouped into a new class of salinity stress since the area of MS and SS was quite limited. Therefore, six classes of stress combinations namely no stress, MD, SD, salinity (MS+SS), MD+MS, and MD+SS were analyzed for the study area.

### 2.2 Traits retrieval

### 2.2.1 Satellite data

The Sentinel-2 mission consists of two satellites equipped with the high-resolution Multispectral Instrument (MSI) in the same orbit. This sensor acquires 13 spectral bands (with varying spatial resolutions) in the visible and near-infrared spectrum at 5 days of revisit times (ESA, 2015). In our study, we used both the 10m and 20m Level 2A observations, downloaded from The Copernicus Open Access Hub (https://scihub.copernicus.eu/), to facilitate the requirement of the Sentinel Application Platform (SNAP) toolbox for both optical and near-infrared observations to be available for determining the functional traits. To create consistency across the bands, those with a 20m resolution (B5, B6, B7, B8A, B11, and B12) were resampled to the 10m resolution of B3 and B4. In total, eight cloud-free scenes were found (21/04/2018, 06/05/2018, 26/05/2018, 30/06/2018, 15/07/2018, 13/09/2018, 13/10/2018, and 28/10/2018) to cover the crop growth cycle. Although additional cloud-free scenes were found in August (04/08/2018, 09/08/2018, 14/08/2018, 19/08/2018, 24/08/2018, and 29/08/2018), none were of high quality, and we therefore choose to omit August from our analysis.

### 2.2.2 Traits selection

Plant traits (e.g. LAI, FAPAR, FVC, Cab and Cw) were selected in consideration of their corresponding impacts on crop functioning and their potential for assessment by remote sensing. LAI is a critical vegetation structural trait related to various plant functioning processes such as primary productivity, photosynthesis, and transpiration (Asner et al., 2003; Boussetta et al., 2012; Fang et al., 2019; Jarlan et al., 2008). FAPAR depends on vegetation structure, energy exchange, and illumination conditions while FAPAR is also an important parameter to assess primary productivity (Liang, 2020; Weiss and Baret, 2016). FVC is a promising parameter corresponding to the energy balance process such as temperature and evapotranspiration (Weiss and Baret, 2016). Cab is an effective indicator of stress and is strongly related to photosynthesis and resource strategy (Croft et al., 2017). Cw plays an important role in transpiration, stomatal conductance, photosynthesis, and respiration (Bowman, 1989; Zhu et al., 2017), as well as in drought assessment (Steidle Neto et al., 2017).

### 2.3 Dataset processing

The biophysical processor within the SNAP toolbox derives the five traits, namely LAI, FAPAR, FVC, canopy chlorophyll content (CCC), and canopy water content (CWC), for each pixel from the Sentinel-2 top of canopy reflectance data at a 10m-resolution for each month. This processor utilizes an artificial neural network (ANN) approach, trained using the PROSAIL simulated database (Weiss and Baret, 2016). This training utilized canopy traits rather than leaf traits (estimated by multiplication with LAI) to improve their neural network performance. To obtain their leaf counterparts (Cw and Cab), to create fully independent variables, CCC and CWC thus need to be divided by LAI to obtain Cab (=CCC / LAI) and Cw (=CWC / LAI). Pixels with quality flags were eliminated from the dataset. It was observed that in April no crop had yet been planted. Instead, we observed that only along the edge of the plots, e.g. in ditches, vegetation was found. This feature was used to generate a ditch map and to mask out pixels in trait maps for the other months. For each variable and each date, only data within the 95% confidence interval were taken to increase data robustness.

### 2.4 Analysis

Since the pixel counts of the six classes of stress combinations namely no stress, MD, SD, salinity, MD+MS, and MD+SS were (highly) different, drought and salinity were not considered as two independent factors. Instead, two-way analysis of variance (ANOVA) was applied to test the main effects and the interactive effect between stress combinations (consisting of 6 levels) and time (5 months) on each individual crop trait. Significant effects of the main stress condition were

investigated through post hoc tests to test whether interaction effects between drought and salinity had occurred. Two-way ANOVAs were run separately for each trait and each crop type (maize and potato) as we expected different patterns. In the Netherlands, potato and maize are planted between mid-April to early May. Crops are surfacing in May and harvested in October. Therefore, to evaluate the response of crops to stresses across the growing season, the effect of drought and salinity on crop traits was determined for May, June, July, and September with a one-way ANOVA. Tukey HSD post hoc tests were performed to identify the differences among the six stress combinations. All statistical analyses were performed with SPSS 27.0 (SPSS Inc., USA).

## 3 Results

### 3.1 Stress impacts depend on the moment in the growing season

The two-way ANOVAs revealed strong effects of date and stress level on the five traits with effect sizes of the response ($R^2$) ranging from 0.326 to 0.796 for the five traits, which was similar for maize and potato. For both maize and potato, $R^2$ values were lowest for Cab and highest for LAI, FAPAR, and FVC. For maize, we found a significant main effect of both date and stress ($p < 0.05$) for Cab, Cw, FAPAR, and FVC. In contrast, LAI was not significantly different across the different stress conditions. For potato, all main effects of date and stress were significant for all five crop traits (Table 1). For all traits and both crops, the interaction between the effects of time and stress conditions was significant ($p < 0.05$) (Table 1), indicating that the impact of stress depended on the moment in the growing season. Despite the significant interaction terms, the partial Eta squared values (Table 1) showed that the effects of time in the growing season were much stronger than those of stress or the interaction of date and stress. The effects of date for maize were stronger than for potato. Interestingly, the effects of the interaction between date and stress were stronger than those of the main effects of stress, suggesting strongly time-specific impacts of stress on the crop traits investigated. The interaction terms were strongest for FVC.

**Table 1.** Two-way ANOVA for different crop traits by time series and stress interactions.

| Crops | Traits | Factors | F | $p$ | Partial Eta Squared | $R^2$ |
|-------|--------|---------|------|-------|---------------------|-------|
| Maize | LAI | date | 2144.5 | 0.000 | 0.636 | 0.766 |
| | | stress | 1.4 | 0.226 | 0.001 | |
| | | date*stress | 8.5 | 0.000 | 0.033 | |
| | Cab | date | 333.9 | 0.000 | 0.222 | 0.326 |
| | | stress | 10.7 | 0.000 | 0.008 | |
| | | date*stress | 3.6 | 0.000 | 0.015 | |
| | Cw | date | 952.1 | 0.000 | 0.449 | 0.590 |
| | | stress | 9.9 | 0.000 | 0.007 | |
| | | date*stress | 4.0 | 0.000 | 0.017 | |
| | FAPAR | date | 1865.9 | 0.005 | 0.603 | 0.738 |
| | | stress | 3.3 | 0.000 | 0.002 | |
| | | date*stress | 8.5 | 0.000 | 0.033 | |
| | FVC | date | 2022.5 | 0.000 | 0.622 | 0.761 |
| | | stress | 22.1 | 0.000 | 0.015 | |
| | | date*stress | 28.7 | 0.000 | 0.105 | |
| Potato | LAI | date | 752.1 | 0.000 | 0.273 | 0.782 |
| | | stress | 13.7 | 0.000 | 0.006 | |
| | | date*stress | 8.1 | 0.000 | 0.020 | |
| | Cab | date | 96.4 | 0.000 | 0.050 | 0.329 |

| | | F | p | Partial Eta Squared | R² |
|---|---|---|---|---|---|
| | stress | 54.2 | 0.000 | 0.024 | |
| | date*stress | 8.7 | 0.000 | 0.023 | |
| Cw | date | 347.4 | 0.000 | 0.158 | 0.571 |
| | stress | 68.1 | 0.000 | 0.030 | |
| | date*stress | 10.3 | 0.000 | 0.027 | |
| FAPAR | date | 612.7 | 0.000 | 0.234 | 0.744 |
| | stress | 25.8 | 0.000 | 0.011 | |
| | date*stress | 14.0 | 0.000 | 0.034 | |
| FVC | date | 844.0 | 0.000 | 0.297 | 0.796 |
| | stress | 18.8 | 0.000 | 0.008 | |
| | date*stress | 13.6 | 0.000 | 0.033 | |

Note: $F$ indicates the test statistic of the $F$-test; $p$ indicates whether the effect is statistically significant in comparison to the significance level ($p < 0.05$); Partial Eta Squared indicates the effect size of different factors; $R^2$ indicates the percentage that the model coincides with the data.

## 3.2 Response of LAI, FAPAR, FVC to drought and salinity

Given the significance of both date and stress and their interactions, subsequent one-way ANOVAs were performed to compare the effects of drought and salinity on LAI, FAPAR, and FVC for maize and potato in May, June, July, and September separately (Fig. 3). The patterns for LAI, FAPAR, and FVC were very similar, although they differ in details and were therefore treated together.

For maize, all of LAI, FAPAR, and FVC obtained their lowest value under MD+SS stress conditions in May. In June, both LAI and FVC dropped the most under salinity stress and it was significantly ($p < 0.05$) different from MD, MD+MS, and MD+SS conditions, but not significantly different from no stress conditions. In contrast, FAPAR also reached its lowest value (under MD+MS stress conditions) in June but had a significant difference ($p < 0.05$) compared with no stress conditions. Both in July and September, LAI, FAPAR, and FVC all had the lowest value under SD conditions, and the difference was significant compared with no stress conditions.

For potato, LAI, FAPAR, and FVC had the lowest ($p < 0.05$) value under MD+MS and MD+SS stress conditions in May. In June, LAI, FAPAR as well as FVC reached the lowest value under SD conditions and were significantly lower than in most other stress conditions even though the difference was not significant from no stress conditions. In July, there was a tendency for LAI, FAPAR, and FVC to be lower under stress conditions, although none of the effects were significant. In September, however, LAI, FAPAR, and FVC significantly decreased under MD, MD+MS, and MD+SS conditions, and the difference was significant compared with no stress conditions. In addition, the difference was not significant among these three stress conditions.

Therefore, both for maize and potato, LAI, FAPAR, and FVC dropped the most under SD stress conditions when they reached their respective maximum value, compared with other stress conditions. At the same time, maize and potato were more sensitive to drought than salinity since no significant change was observed between drought conditions and conditions with a combination of drought and salinity stress.

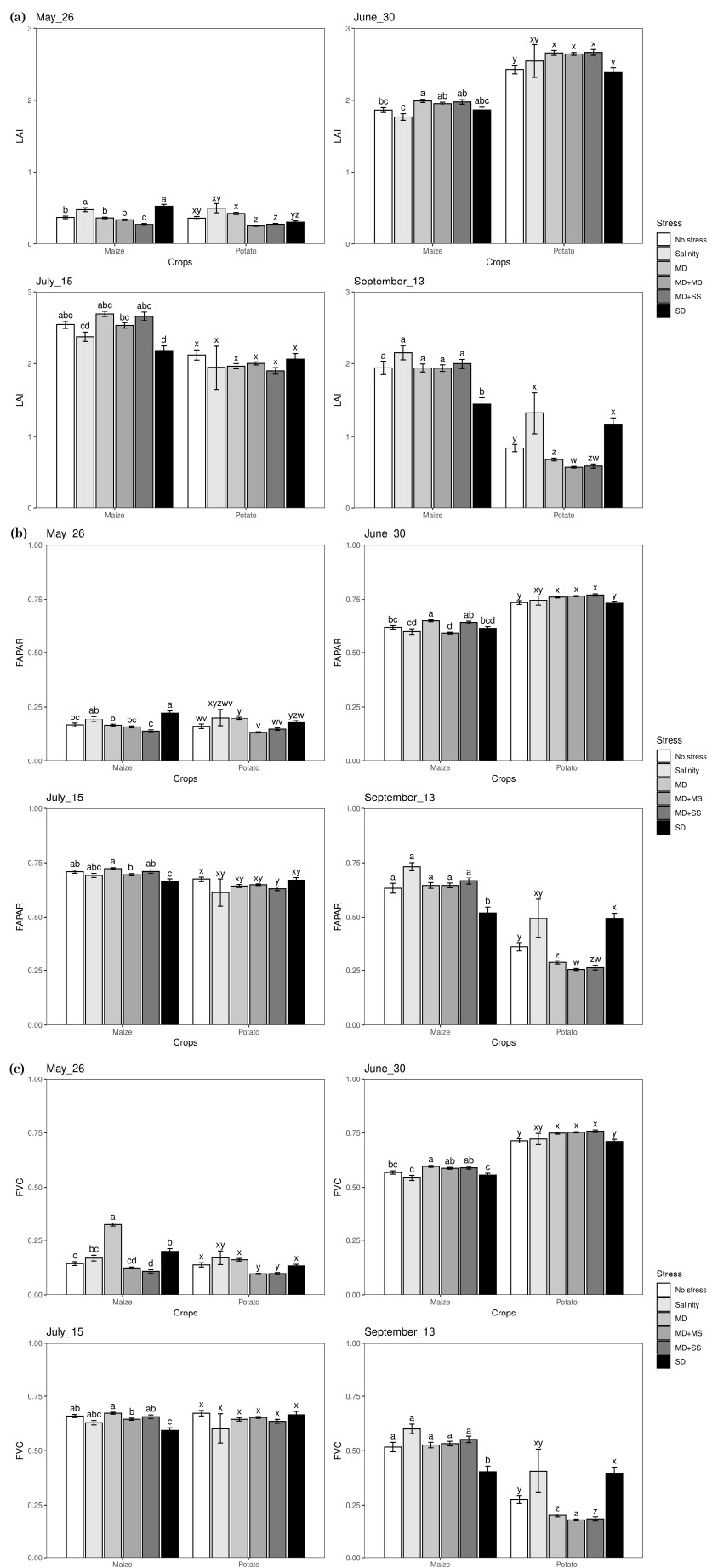

214

 **Figure 3.** Expressions of LAI, FAPAR, and FVC under various stress conditions in May, June, July, and September 2018. Different letters in each panel indicate significant differences ($p < 0.05$). MD, moderate drought only; Salinity, salinity only; MD+MS, moderate drought and moderate salinity; MD+SS, moderate drought and severe salinity (MD+SS); SD, severe drought only.

### 3.3 Response of leaf chlorophyll and water content to drought and salinity

The one-way ANOVAs revealed that there were significant ($p < 0.05$) impacts of the various stress conditions on Cab and Cw (Fig. 4). For maize, Cab obtained its lowest value under salinity stress in May and June while it was not significantly different from no stress conditions. However, in July, Cab reached the lowest value under MD+MS conditions although the difference was not significant from other stress conditions. There were no significant changes observed for Cab in September. For potato, Cab dropped the most under salinity conditions in May although the difference was not significant from no stress conditions. Furthermore, Cab significantly decreased under MD+SS conditions in June and July, compared with other conditions. Although Cab dropped the most under salinity conditions in September, the difference was not significantly different from other conditions. In addition, compared with no stress, potato had the lowest Cab under MD+SS conditions while there was no significant difference between MD+SS and salinity conditions in most growing periods.

Cw decreased under all stress conditions in May, June, and July for both maize and potato, except for SD conditions in May, compared with no stress conditions. At the same time, Cw reached its lowest value under MD+SS conditions and it was significantly different from under no stress conditions. Nonetheless, there were different changes for maize and potato in September. Cw was not significantly different among any conditions for maize while it was the lowest under salinity conditions for potato.

Therefore, this analysis illustrates that salinity affected maize less than drought since crop responses were more obvious to drought than salinity for Cw. In contrast, salinity showed a more severe effect on maize and potato at the early growth stages for Cab. Meanwhile, Cab was affected by co-occurring drought and salinity in June and July for potato. It seems that there was a non-additive effect of drought and salinity for Cw since the changes were not significant between MD+MS, MD+SS, MD, and salinity conditions.

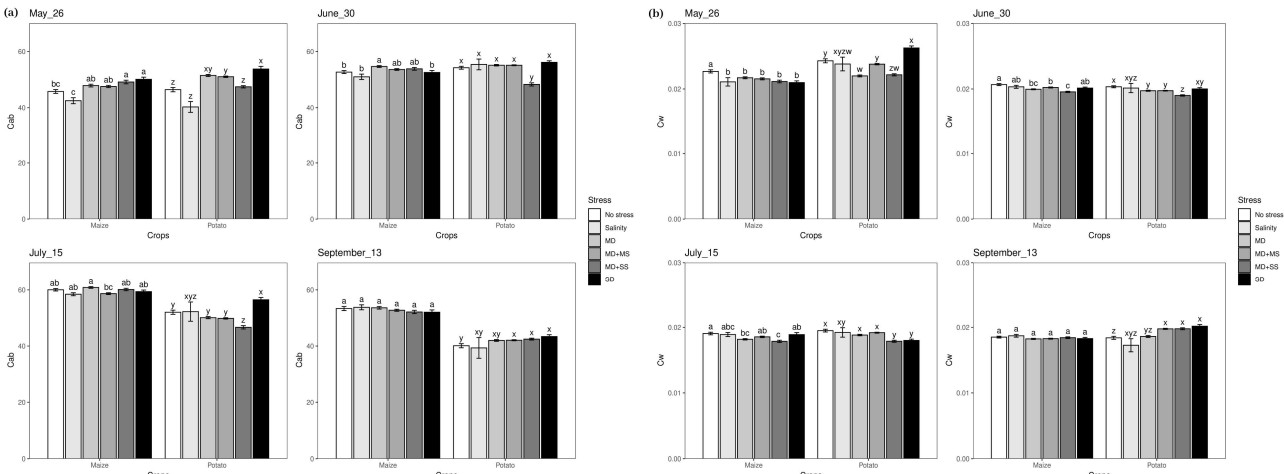

**Figure 4.** Expressions of Cab and Cw under various stress conditions in May, June, July, and September 2018. Different letters in each panel indicate significant differences (p < 0.05). MD, moderate drought only; Salinity, salinity only; MD+MS, moderate drought and moderate salinity; MD+SS, moderate drought and severe salinity (MD+SS); SD, severe drought only.

**4 Discussion**

In this study, we quantified the large-scale impacts of co-occurring drought and salinity on a variety of crop traits using satellite remote sensing. We observed that –in contrast to our expectations – the impacts of salinity were not highly pronounced at this scale, with most strong impacts originating due to drought stress during the 2018 drought. At specific moments in the growing season, salinity and/or the combined effects of salinity and drought pronouncedly affected individual crop traits. In this way, with increasing salinity driven by more intensive droughts, water allocation should not only be governed by the amount of water shortage, but also the salinity of the remaining water. In this paper, we provide the first evidence that those impacts can be monitored through remote sensing. This might provide a basis towards a monitoring system for multiple crops with multiple stresses as well as better governance policies to release this problem.

**4.1 Drought stress is more important than salinity stress in farmers' conditions**

The exacerbating effects of co-occurrent drought and salinity (Fig. 3 and Fig. 4) that we found are consistent with findings of small-scale experiments (e.g. greenhouses). Consistent with our results, synergistic effects of co-occurring water stress and salinity stress have been found on maize reproductive growth and grain formation in a field study (Liao et al., 2022). Spinach (*Spinaciaoleracea* L., cv. Racoon) yield decreased more under co-occurring water-salinity stress in comparison with separate water stress and salinity (Ors and Suarez, 2017). The co-occurring drought and salinity stress was more harmful to cotton root growth compared to their individual effects (Zhang et al., 2013). Moreover, the combined negative effect of drought and salinity stress on *Panicum antidotale* was stronger than that of single stress (Hussain et al., 2020). Our research showed that the outcomes of these small-scale experimental studies also apply to real large-scale environments, where different sources of variance are present. Specifically, we show that in real farmers' conditions, the co-occurrence of drought and salinity indeed can constitute a severe threat due to its interactive effects on crop growth.

In addition, we evaluated whether drought or salinity stress has more impact on crop performance. We observed that maize and potato were generally more sensitive to drought than salinity in this study (Fig. 3 and Fig. 4). This is consistent with results of previous studies that highlight that drought impacts are generally more detrimental than salinity stress for crops, e.g. for sesame (*Sesamum indicum*) (Harfi et al., 2016), *Mentha pulegium* L. (Azad et al., 2021), durum wheat (Sayar et al., 2010), grass pea (Tokarz et al., 2020), and sweet sorghum (Patane et al., 2013). However, given that the threshold of salinity at which crop damage occurs (according to the FAO guidelines (Ayers and Westcot, 1985)) was surpassed in all situations in which salinity stress was imposed (including in our study), we initially expected salinity to be a stronger explanatory variable than drought. As such, salinity impacts on crop performance (by the FAO) may have been overestimated. Indeed, in an experimental field situation in which drought stress was carefully avoided, higher thresholds of salinity-induced damage were observed for potato (van Straten et al., 2021).

In combination, the results from our study (supported by results from other studies) suggest that salinity particularly induces adverse effects when co-occurring with drought stress. Water stress impacts on photosynthesis and biomass of plants were extenuated by salinity since salinity enhances the synthesis of ATP and NADPH by promoting photosynthetic pigments and photosystem II efficiency. The impacts of combined drought and salinity stress on plant growth, chlorophyll content, water use efficiency, and photosynthesis were less severe compared to drought alone. This indicates compensating effects on carbon assimilation due to osmotic adjustments induced by $Na^+$ and $Cl^-$ (Hussain et al., 2020). Thus, the detrimental effect of single drought stress on crop growth is considered to be mitigated by salinity.

## 4.2 Drought and salinity stress differ between growth stages

The responses to drought and salinity stress were different at different growth stages of the crops. This was expressed by the significant interactions between the effects of time and stress conditions for all of our crop responses (Table 1). We found that during the grain filling (maize) and tuber bulking phase (potato), the sensitivities of these crops are expressed distinctly in the non-harvested aboveground tissues (Fig. 3 and Fig.4), with clear differences in the remote sensing plant traits.

Given that we were not able to monitor the harvestable products, multiple mechanisms may explain these patterns. The relatively high leaf coverage (as related to LAI, FAPAR, and FVC) at salinity and severe drought conditions at the end of the growing season may be an expression of a compensation process. Specifically, early and prolonged drought could have led to more assimilates allocated to non-harvestable potato parts for drought resistance since the number of tubers reduced (Jefferies, 1995; Schittenhelm et al., 2006). In that case, we should consider their higher leaf coverage at the end of the season as a survival mechanism, rather than true drought tolerance, leading to reduced tuber yields (Daryanto et al., 2016b). Future studies that combine remote sensing with harvesting data may be able to evaluate this mechanism in more detail.

In our study, different response patterns of maize and potato occurred to the different stresses over the growing season. This is consistent with previous studies focusing on the impact of drought and/or salinity onsets. For potato, it has been suggested that tuber yields particularly decreased when drought stress occurs during the vegetative and tuber initiation stages than during the tuber bulking stage (Wagg et al., 2021), although another study observed the reverse pattern (Daryanto et al., 2016b). For maize, on the other hand, drought seems to have the most detrimental impact during the maturation stage (Mi et al., 2018; Zhang et al., 2019), and the reproductive phase (Daryanto et al., 2016a; Daryanto et al., 2017). Considering the additional co-varying factors within our 'real-life' study, it is very probable that we were able to detect similar effects. This suggests that we may use satellite remote sensing –albeit less spatially precise than e.g. sensing through drones- as a cost-effective early warning signal for detecting drought and salinity stress at moments during the growing season when differences in crop performance are still subtle.

## 4.3 Crop responses to stress can be better understood with a multi-trait approach

In addition to facilitating the evaluation of crop performance during multiple stages of the growing season (in contrast to most destructive methods), remote sensing also allows a multi-trait approach to better understand the mechanisms involved in crop responses. Each of the five traits is associated with different functions of plants that might be individually impacted by the different stresses. Therefore, focusing on only one individual metric (as commonly done; see Wen et al. (2020) for a review) limits our capacity to gain full insight into drought and salinity responses. Hence, given that individual crop traits may respond differently to drought and salinity reflecting its stress resistance and tolerance strategy, the evaluation of these distinct responses may help to understand this strategy.

In this study, Cw was consistently lower in all drought and salinity treatments as compared to no stress conditions in May, June, and July. Indeed, this is a common response of plants in response to drought and salinity (e.g. Wen et al., 2020). In this respect, it is interesting that no decrease in Cw was observed at the end of the growing season, in September. Whether the phenomenon is related to the survival mechanism mentioned above or to the lower transpiration demands at the end of the season because of lower aboveground biomass, cannot be concluded from these data. Some evidence pointing to the survival mechanism is the finding (Ghosh et al., 2001; Levy, 1992) that the leaf dry matter increased for potato under drought/salinity stress (like in our study) while the dry matter of the tubers appeared to have a greater decline.

With respect to chlorophyll contents, we observed a decline in Cab under salinity conditions in May and the MS+SS treatment in June and July, while no decrease was observed in any of the treatments exposed to drought only. This indicates that while total leaf area was not (much) affected by salinity, the salinity did negatively affect crop performance. It has been reported that chlorophyll content in maize was significantly reduced upon salinity, along with other plant traits including plant height, shoot/root biomass, and leaf numbers (Fatima et al., 2021; Mahmood et al., 2021). Likewise, similar patterns were obtained in potato plants in saline soil (Efimova et al., 2018). Hence, this implies that soil salinity tends to negatively affect crop growth and restrict nutrient uptake.

Cab and Cw responses to drought and salinity were distinct from responses of LAI, FAPAR, and FVC (Fig. 3 and Fig. 4). LAI, FAPAR, and FVC showed similar patterns to stress due to their highly physical correlation (Hu et al., 2020). The different patterns of Cw and Cab point to different drought and salinity resistance strategy components associated with these traits: LAI (and FAPAR/FVC) reflect the decrease in biomass due to stress, partly because stress directly and negatively impacts growth and partly because having lower biomass decreases the evapotranspiration demands of the crop, which increases the resilience of the crop to deal with drought. Cw represents another pathway to reduce evapotranspiration demands, i.e. by reducing the amount of water per gram of leaves. Also, this response may be a direct effect of the more negative pressure heads due to drought or due to increased osmotic pressures (due to salinity). It may also be part of the adaptive strategy of the crop to increase its resilience. Cab also responds to drought and salinity, but in its own way, i.e. by adapting its photosynthetic capacity while being affected by a lower stomatal conductance (due to drought and/or salinity). See e.g. Wright et al. (2003) for a framework explaining these nitrogen-water interactions.

In addition, our approach gives the insight to analyze the effect of stresses on yield based on the five traits, even though yield cannot directly be derived from remote sensing. Traits including Cab, LAI, and FAPAR, have been used (either separately or in combination) as a proxy for final yield estimates from remote sensing in many studies. For instance, NDVI -which is based on the combination of LAI and Cab- is extensively used to estimate crop yield (Huang et al., 2014; Mkhabela et al., 2011; Vannoppen et al., 2020). Also, LAI itself has been used for predicting the final yield (Dente et al., 2008; Doraiswamy et al., 2005; Sun et al., 2017). Meanwhile, Cab and FAPAR were also proven to be highly correlated with crop yield (Ghimire et al., 2015; López-Lozano et al., 2015). Thus, while yield cannot be estimated directly from remote sensing or ground truth data at the desired high spatial resolution, our indicators do relate to yield and can be used in more application-based contexts to inform on yield impacts.

**4.4 Implications for future research and management.**

The number of studies that evaluate the effects of drought and salinity stress on crops is limited (Wen et al., 2020). In general, studies focus on small-scale experimental studies under strictly control of all variables with only a limited number of crops (Hussain et al., 2020; Ors and Suarez, 2017). To our knowledge, this is the first study that uses satellite remote sensing to investigate drought and salinity impacts for a large area under real-life conditions necessary for constructing stress management policies.

In such real-life conditions, as investigated here, irrigation of crops is commonly applied as management practice during drought events to reduce the severity of drought impacts (Deb et al., 2022; Lu et al., 2020). In this study, however, we have evidence that irrigation did not play a major role in the patterns found since all croplands included in our research area were identified as rainfed cropland (according to the ESA/CCI land cover map in 2018; https://maps.elie.ucl.ac.be/CCI/viewer/). In addition, while farmers in the area are known to irrigate their cropland, the Dutch government announced a temporary national irrigation ban in 2018 (for various areas including our research area)

to spare water (Perry de Louw, 2020). As a consequence, we could not analyze the impacts of irrigation management on
the combined effects of drought and salinity. This might potentially be solved by investigating other drought historic events
with moderate severity in Europe, such as the year of 2003 (Ciais et al., 2005) or 2015 (Ionita et al., 2017) in Europe, when
such a ban was not executed. Unfortunately, satellite remote sensing observations with the required 20-30m resolutions of
these events are limited, as Sentinel-2 was only launched in 2015 and the Landsat satellites provide a too coarse temporal
resolution.
Likewise, impacts of salinity and drought are moderated by crop selection. Traditionally, farmers do not plant highly
vulnerable crops in moderate/high salinity areas. In fact, we found crops sensitive to salinity such as apple (Ivanov, 1970)
and broccoli (Bernstein and Ayers, 1949) to be abundant in non-saline areas but only little in saline areas. To ensure an
accurate evaluation of salinity impacts, we only investigated those crops with a significant abundance in all available stress
conditions. More sensitive crops might even respond more strongly.
**5 Conclusions**
In this study, we present the first attempt to evaluate the real-life effects of drought, salinity, and their combination on crop
health using multiple traits from remote sensing monitoring during 2018 over the Netherlands. Our approach gives new
insights for monitoring crop growth under co-occurring stresses at a large scale with high-resolution data. We found that
while in general temporal patterns –reflecting crop growth dynamics- were stronger than effects of stress conditions, stress
impacts depended on the time of the growing season. Furthermore, we also found that the temporal dynamics in crop
responses to drought and salinity were different for maize vs. potato. In general, the five investigated traits were more
negatively affected by a combination of drought and salinity stress compared to individual stress. Meanwhile, both maize
and potato responded more prominently to drought, thus demonstrating a stronger sensitivity, than to salinity. Specifically,
LAI, FAPAR, and FVC dropped the most under severe drought stress conditions. Consequently, the proposed new
approach poses a facilitated way for simultaneously monitoring the effect of drought and salinity on crops in large-scale
agricultural applications.

*Data availability*. The drought map of the Netherlands in 2018 is retrieved from Chen et al. (2022). The top-soil salinity
map of the Netherlands is retrieved from The Netherlands Hydrological Instrumentarium (NHI) (https://data.nhi.nu/). The
crop map of the Netherlands in 2018 is retrieved from the Key Register of Parcels (BRP) of the Netherlands Enterprise
Agency (https://www.pdok.nl/introductie/-/article/basisregistratie-gewaspercelen-brp-). All satellite scenes are
downloaded from The Copernicus Open Access Hub (https://scihub.copernicus.eu/). The dataset relevant to this study is
available upon request from the corresponding author.

*Author contributions*. Conceptualization, JT, PVB, and WW; methodology, JT, QC, WW, and PVB.; investigation, WW
and QC; writing—original draft preparation, WW; writing—review and editing, PVB. and JT; supervision, PVB, and JT
All authors have read and agreed to the published version of the manuscript.

*Competing interests*. The authors declare no conflict of interest.

*Financial support*.  This work was supported by the China Scholarship Council (CSC).

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
