# Peer review of "Monitoring the combined effects of drought and salinity stress on crops using remote sensing in the Netherlands"

_Hydrology and Earth System Sciences, 2022_

## Author Comment (AC1)

**Response to Reviewer 1**

**General comment**

The combined effect of drought and salinity on crops is very important for food security under global change background. Remote sensing shows its advantage for large-scale applications. This paper used the sentinel-2 satellite data to conduct an analysis in this regard. The findings are interesting.

**Response:** Thank you for your positive comments of our work. We appreciate that our work provides insights into monitoring drought and salinity impacts on crops by remote sensing on a large scale. Our itemized responses are attached below. In order to facilitate the review, the comment of the reviewer is displayed in black, and the reply is displayed in blue font.

**Major concerns**

Q1. Irrigation can mitigate the drought effect to a large extent. I would like to know how irrigation has influenced the analysis. There is no information reported in this regard.

**Response:** We agree that irrigation plays an important role in relieving drought impacts on crops in many cases. In this study, we have evidence that irrigation did not play a major role in the patterns found. First of all, all croplands included within our research area have been identified as rainfed cropland (see Fig. 1 below) according to the ESA/CCI land cover map in 2018 (https://maps.elie.ucl.ac.be/CCI/viewer/). In addition, while -in the area- farmers are known to irrigate their cropland, the Dutch government announced a temporary national irrigation ban in various areas including our research area in 2018 (Perry de Louw, 2020) to spare water. Therefore, we assumed that irrigation management was absent during our study period. We agree that in other cases, irrigation may modify several of the patterns found and we will add a remark on this to the discussion section of the revised manuscript.

[Figure]

**Land cover**

| | | | |
|---|---|---|---|
| ☐ | Water bodies | ▓ | Mosaic natural vegetation |
| ▨ | Bare areas | ▨ | Mosaic tree and shrub/herbaceous cover |
| ▨ | Cropland, rainfed | ▨ | Shrub/herbaceous cover |
| ▨ | Grassland | ▨ | Sparse vegetation |
| ▨ | Mosaic cropland | ▨ | Tree cover |
| ▨ | Mosaic herbaceous cover/tree and shrub | ▨ | Urban areas |

Fig. 1 The land cover map of North-Holland

Q2. Five different indicators were used to depict the health condition of different crops. I am wondering how the stress factors influence the final yield. Is it possible to have some discussion in this regard.

**Response:** Yield information is not available at pixel level but only at farm level in the Netherlands. Particularly, given the average size of a farm, such information is not insightful at the scale with which we are working. Moreover, there are no yield products available based on remote sensing at such high resolution. Therefore, we cannot directly derive yield estimates from remote sensing and we used these five crop traits instead to understand crop responses. Traits including Cab, LAI, and FAPAR, have been used (either separately or in combination) as proxy for final yield estimates from remote sensing in many studies. For instance, the Normalized Difference Vegetation Index (NDVI) which is based on the combination of LAI and Cab, is extensively used to estimate crop yield (Huang et al., 2014;Mkhabela et al., 2011;Vannoppen et al., 2020). Also LAI itself has been used for predicting final yield (Sun et al., 2017;Dente et al., 2008;Doraiswamy et al., 2005). Meanwhile, Cab and FAPAR were also proven to be highly correlated with crop yield (López-Lozano et al., 2015;Ghimire et al., 2015). Thus, while yield cannot be estimated directly from remote sensing or ground truth data at the desired high spatial resolution, our indicators do relate to yield and can be used in more

application-based contexts to inform on yield impacts. In our revised manuscript, we will highlight these relationships with the final yield to make the link to wider application of our findings.

**Minor comments**

Q3. Line 33, more deeply challenged.

**Response:** We will revise "deeper challenged" to "more deeply challenged".

Q4. Line 37, delete 'of' and 'more than'

**Response:** We will delete 'of' and 'more than' from this sentence.

Q5. Lines 83-90, why was SPEI selected as the drought indicator rather than the others? What is the RD_new projection? Where are the precipitation and PET data from?

**Response:** There are several common drought indices including the Palmer Drought Severity Index (PDSI), the Standardized Precipitation Index (SPI), the Standardised Precipitation-Evapotranspiration Index (SPEI), etc., to evaluate drought events. PDSI, which is based on the water balance equation, has disadvantages due to autoregressive characteristics and its fixed temporal scale (Guttman, 1998). SPI, which is calculated from precipitation data, shows better performance than PDSI on droughts detection thanks to its multi-scalar features (Hayes et al., 1999). Nevertheless, compared with these two common drought indices, the SPEI is a multiscale drought index based on precipitation and temperature data, and in this way, it has the advantage of detecting, monitoring, and assessing drought in multiple systems (Vicente-Serrano et al., 2010).

RD_new (EPSG:28992) projection is a projected coordinate reference system of the Netherlands. All maps were projected to RD_new projection to create consistent data layers. We will explain this in the revised manuscript.

The drought map was created by our group, and published in another study in the journal Science of the Total Environment (Chen et al., 2022). The precipitation data were fused based on remote sensing data and ground observations. The PET data was obtained from MODIS in 8-day composite dataset. More details can be found in Chen et al. (2022).

Q6. Lines 124-129, Include some information about Sentinel-2 in the data description although it was pointed out in Fig. 1.

**Response:** The following information about Sentinel-2 will be added to the revised manuscript:

The Sentinel-2 mission consists of two satellites equipped with high-resolution Multispectral Instrument (MSI) in the same orbit. The satellites acquire 13 spectral bands from the visible spectrum to the short-wavelength infrared spectrum in 5 days revisit times at a spatial resolution of 10m, 20m, and 60m (ESA, 2015). In our study, we used the 10m resolution as the SNAP toolbox requires both optical and near-infrared observations to be available for determining the functional traits. Bands in 20m including B5, B6, B7, B8A, B11 and B12 were resampled to 10m resolution to match consistency with B3 and B4.

Q7. Line 145, why was the biomass effect removed? Is this contradictory to the Cab*LAI and Cw*LAI at Line 142?

**Response:** In order to use truly independent variables within our analysis, we removed the biomass effect: SNAP uses a neural network to derive five canopy traits, namely LAI, FAPAR, FVC, canopy chlorophyll content (CCC), and canopy water content (CWC) (lines 141-142). However, these canopy traits are internally (in SNAP) calculated as (Cab*LAI) and (Cw*LAI), i.e. based on their leaf equivalents, Cab and Cw. Since both CCC and CWC are calculated from LAI, we divided them by LAI to obtain their leaf counterparts (Cw and Cab) to create fully independent variables. Thus, with the removal of the biomass effect, we mean the removal of the effects of LAI –which is also calculated separately within SNAP- from our functional trait estimates. We will modify the text to clarify this.

Q8. Line 151, What do you mean by 'due to the unbalance in the occurrence of stress conditions'?

**Response:** The unbalance in the occurrence of stress conditions means that the pixel counts of the six classes of stress combinations namely no stress, MD, SD, salinity (MS+SS), MD+MS, and MD+SS were (strongly) different. We will clarify this in the revised manuscript.

Q9. Lines 163-173, More explanations are needed to illustrate the connotations of different indicators in the ANOVA analysis, to increase the readability. Probably this can be supplemented in the methodology section.

**Response:** We will add the following information on the different indicators in the notes of Table 1 to make it more clear for readers:

Notes: *F* indicates the test statistic of the F-test; *p* indicates whether the effect is statistically significant in comparison to the significance level ($p < 0.05$); Partial Eta Squared indicates the effect size of different factors; $R^2$ indicates the percentage that the model coincides with the data.

Q10. Line 200, Add some information for the different letters indicating the significance level.

**Response:** All the significance levels are $< 0.05$ in Fig. 3 and Fig. 4. The letters in Fig. 3 and Fig. 4 indicate whether there is a significant difference among different stress groups based on the pairwise comparison. If the letter in one group is different from the other group, then a significant difference exists between these two groups. We'd like to clarify that it is a common way to show significant differences in bar plots throughout scientific literature as we have explained with the sentence 'Different letters in each panel indicate significant differences ($p < 0.05$)' in the caption.

Q11. Line 220 and Line 244, It was concluded at Line 220 that there is no additive effect for drought and salinity. Is it in contradiction to the severe effect of the co-occurrence of drought and salinity?

**Response:** In fact, these two points are not contradicting. The effects of two factors can be either additive or interactive. If two factors are additive, then the effect of both factors (in this case drought and salinity) equals the sum of the effects of the individual factors. Thus, the reduction in a trait value by the combination of drought and salinity would equal the reduction due to drought plus the reduction due to salinity. If the effects of two factors are interactive,

then the combined effect of two co-occurring factors does not equal the sum of the individual effects. That was the case here in which we found that particularly the MD+SS conditions led to major impacts on our functional traits. We will revise our statements to make this distinction clear.

Q12. Line 257, why was the drought effect mitigated? Please add more explanations.

**Response:** More explanations will be added to the revised manuscript as follows:

Water stress impacts on photosynthesis and biomass of plants were extenuated by salinity since salinity enhances the synthesis of ATP and NADPH by promoting photosynthetic pigments and photosystem II efficiency. The impacts of combined drought and salinity stress on plant growth, chlorophyll content, water use efficiency, and photosynthesis were less severe compared to drought alone. This indicates the compensating effects on carbon assimilation due to osmotic adjustments induced by $Na^+$ and $Cl^-$ (Hussain et al., 2020). Thus, the detrimental effect of single drought stress on crop growth is considered to be mitigated by salinity.

Q13. Line 278, 'Considering the additional', remove the comma. Change 'promising' to 'probable'.

**Response:** We will delete ',' and change 'promising' to 'probable'.

Q14. Line 283, 'In addition to facilitating the evaluation…'

**Response:** We will change 'being able to evaluate' to 'facilitating the evaluation'.

Q15. Line 285, distinctively

**Response:** We will revise this sentence to make it easier to understand. This sentence will be revised to: 'In our study, Cab and Cw responses to drought and salinity were distinct from responses of LAI, FAPAR, and FVC (Fig. 3 and Fig. 4).'.

Q16. Line 288, understand

**Response:** We will revise this sentence to: 'Given that individual crop traits may differently respond to drought and salinity, reflecting their stress resistance and tolerance strategies, the evaluation of these distinct responses may help to understand these strategies.'.

Q17. Line 289, as compared to

**Response:** We will revise this sentence to: 'In this study, Cw was consistently lower in all drought and salinity treatments as compared to no stress conditions in May, June and July.'.

Q18. Line 291, In this respect

**Response:** We will revise 'that' to 'this' in the sentence.

Q19. Line 292, the transpiration demand normally refers to the atmospheric demand, like VPD and incoming radiation. What do you mean here?

**Response:** Here transpiration means the loss of leaf water vapor.

Q20. Please check the English writing more carefully to enhance the readability.

**Response:** We will carefully revise the manuscript in terms of English writing to improve the readability.

**References**

Chen, Q., Timmermans, J., Wen, W., and van Bodegom, P. M.: A multi-metric assessment of drought vulnerability across different vegetation types using high-resolution remote sensing, Sci. Total Environ., 154970, https://doi.org/10.1016/j.scitotenv.2022.154970, 2022.

Dente, L., Satalino, G., Mattia, F., and Rinaldi, M.: Assimilation of leaf area index derived from asar and meris data into ceres-wheat model to map wheat yield, Remote Sens. Environ., 112, 1395-1407, https://doi.org/10.1016/j.rse.2007.05.023, 2008.

Doraiswamy, P. C., Sinclair, T. R., Hollinger, S., Akhmedov, B., Stern, A., and Prueger, J.: Application of modis derived parameters for regional crop yield assessment, Remote Sens. Environ., 97, 192-202, https://doi.org/10.1016/j.rse.2005.03.015, 2005.

ESA: Sentinel-2 user handbook, https://sentinel.esa.int/documents/247904/685211/sentinel-2_user_handbook, 2015.

Ghimire, B., Timsina, D., and Nepal, J.: Analysis of chlorophyll content and its correlation with yield attributing traits on early varieties of maize (zea mays l.), J. Maize Res. Dev., 1, 134-145, https://doi.org/10.3126/jmrd.v1i1.14251, 2015.

Guttman, N. B.: Comparing the palmer drought index and the standardized precipitation index, J. Am. Water Resour. Assoc., 34, 113-121, https://doi.org/10.1111/j.1752-1688.1998.tb05964.x, 1998.

Hayes, M. J., Svoboda, M. D., Wiihite, D. A., and Vanyarkho, O. V.: Monitoring the 1996 drought using the standardized precipitation index, Bull. Am. Meteorol. Soc., 80, 429-438, https://doi.org/10.1175/1520-0477(1999)080<0429:MTDUTS>2.0.CO;2, 1999.

Huang, J., Wang, H., Dai, Q., and Han, D.: Analysis of ndvi data for crop identification and yield estimation, IEEE J. Sel. Top. Appl. Earth Obs. Remote Sens., 7, 4374-4384, https://doi.org/10.1109/JSTARS.2014.2334332, 2014.

Hussain, T., Koyro, H. W., Zhang, W., Liu, X., Gul, B., and Liu, X.: Low salinity improves photosynthetic performance in panicum antidotale under drought stress, Front. Plant Sci., 11, 481, https://doi.org/10.3389/fpls.2020.00481, 2020.

López-Lozano, R., Duveiller, G., Seguini, L., Meroni, M., García-Condado, S., Hooker, J., Leo, O., and Baruth, B.: Towards regional grain yield forecasting with 1km-resolution eo biophysical products: Strengths and limitations at pan-european level, Agric. For. Meteorol., 206, 12-32, https://doi.org/10.1016/j.agrformet.2015.02.021, 2015.

Mkhabela, M. S., Bullock, P., Raj, S., Wang, S., and Yang, Y.: Crop yield forecasting on the canadian prairies using modis ndvi data, Agric. For. Meteorol., 151, 385-393, https://doi.org/10.1016/j.agrformet.2010.11.012, 2011.

Perry de Louw, V. K., Harry Massop, Ab Veldhuizen Beregening: Deltafact, Alterra - Soil, water and land use, Amersfoort 2020.

Sun, L., Gao, F., Anderson, M. C., Kustas, W. P., Alsina, M. M., Sanchez, L., Sams, B., McKee, L., Dulaney, W., White, W. A., Alfieri, J. G., Prueger, J. H., Melton, F., and Post, K.: Daily mapping of 30 m lai and ndvi for grape yield prediction in california vineyards, Remote Sens., 9, https://doi.org/10.3390/rs9040317, 2017.

Vannoppen, A., Gobin, A., Kotova, L., Top, S., De Cruz, L., Vīksna, A., Aniskevich, S., Bobylev, L., Buntemeyer, L., Caluwaerts, S., De Troch, R., Gnatiuk, N., Hamdi, R., Reca Remedio, A., Sakalli, A., Van De Vyver, H., Van Schaeybroeck, B., and Termonia, P.: Wheat yield estimation from ndvi and regional climate models in latvia, Remote Sens., 12, https://doi.org/10.3390/rs12142206, 2020.

Vicente-Serrano, S. M., Beguería, S., and López-Moreno, J. I.: A multiscalar drought index sensitive to global warming: The standardized precipitation evapotranspiration index, J. Clim., 23, 1696-1718, https://doi.org/10.1175/2009JCLI2909.1, 2010.

---

## Author Comment (AC2)

**Response to Reviewer 2**

**General comment**

Drought and salinity are considered to be the two main factors limiting crop productivity. Remote sensing enables the assessment of the impacts of extremes on crops, but it is seldomly used in the study of compound effects of drought and salinity stress. The novelty of this study is to assess the impacts of drought, salinity, and their combination on crop traits using multiple remote sensing observations and explore their relationships with stress timings and drought levels. The manuscript makes a contribution to the assessment of compound extremes' impacts using remote sensing and the writing is well organized. I suggest this manuscript should be accepted by HESS after minor revisions.

**Response:** Thank you for your positive comments about our work. We appreciate your support of the novelty and potential applications of compound extremes' impacts on a large scale. Our itemized responses are attached below. In order to facilitate the review, the comment of the reviewer is displayed in black, and the reply is displayed in blue font.

**Specific comments:**

Q1. In this study, only the 2018 case over the Netherlands was analyzed, are the conclusions robust? As the available data range from 2004 to 2018, are there any else cases to verify the conclusions? If there is no more cases, it is better to add the name of this case in the title.

**Response/Action:** From a detailed and thorough review (Wen et al., 2020), we found that at present nobody has investigated the combined effects of drought and salinity at such a scale using remote sensing. So, while this is an important novelty of our study, it also hampers verifying our conclusions with other cases. We use the years 2004 to 2018 to create a baseline of local hydrological conditions and to be able to evaluate the local deviations in those conditions for 2018 (given that 2018 was by far the year with the most extreme drought conditions in that period), but we cannot use the other years to verify our patterns as this would lead to circular arguments. At the same time, we believe our conclusions are robust in light of the approach we used and its consistency with results from small-scale local studies. We are currently validating our analysis for a much larger area in the USA, which should provide extra support for this study. The findings however are outside the scope of this particular paper. Concerning the current limits of this case, we agree with your opinion and will revise the title to 'Monitoring the combined effects of drought and salinity stress on crops using remote sensing in the Netherlands' in the revised manuscript.

Q2. Add a map of the crop distribution over the study region.

**Response/Action:** We will add a crop map of North-Holland province to Figure 2.

Q3. Line 89-90: The standard deviation of SPEI is 1 (Vicente-Serrano et al 2010), why do you define drought when SPEI is less than -321?

**Response/Action:** In this study, we adopted daily SPEI for the period from April 1st to October 30th, a total of 214 days, as this coincided with the crop growth period. We defined -1 and -1.5 as daily thresholds for different drought severity classes according to previous classifications (McKee et al., 1993;Tao et al., 2014). Thus, (cumulative) SPEI for no drought should be

between -214 to 0, SPEI for moderate drought should be between -321 to -214 and for severe drought, SPEI should be lower than -321 for the whole growing period. We will add more information on thresholds of drought classifications to the revised manuscript.

Q4. The captions of Table 1, figure 3, and figure 4 should be described in detail, e.g. what are MD, MS, SS, ab, and abc short for in fig. 3?

**Response/Action:** We will describe the meanings of all stress conditions in the caption of Fig.3 and Fig. 4 of the revised manuscript. All the different letters (e.g. ab) refer to the post hoc result from pairwise comparisons. It is a common way to show significant differences in bar plots throughout scientific literature as we have explained with the sentence 'Different letters in each panel indicate significant differences ($p < 0.05$)' in the caption.

Q5. As figures 3-4 show the values of crop traits from May to September, it is better to show their standardized anomalies compared with climatology, which enables the comparison between different timings.

**Response/Action:** Unfortunately, due to several limitations, we cannot provide a climatology. In this case, we calculated traits based on Sentinel-2 which is only active since 2015. As such, sentinel-2 does not have observations (yet) to investigate the long-term behavior of those traits, thereby limiting us to define a climatology for no-stress conditions. While Landsat and other comparable satellites exist that could be used for this, we chose not to use this information because of the differences in wavelengths and resolutions among different satellites. In addition, we chose not to adopt a non-drought year (like 2015 or 2016) as a control (no stress condition) in this research, because we could not quantify the natural variation (in respect to a long-term climatology) based on those specific years. Hence, we were restricted in our approach to make a comparison to a control treatment. This comparison to the control provides the baseline for no-stress conditions that can be directly compared to the responses in stressed conditions. This avoids the need for including climatology.

Q6. What is the best explanation of the different responses among the five crop traits to such stresses?

**Response/Action:** Each of the five traits is associated with different functions of plants that might be individually impacted by the different stressors. Therefore, focusing on only one individual metric (as commonly done; see Wen et al. (2020) for a review) limits our capacity to gain full insight into drought and salinity responses. Hence we chose explicitly not to focus on analyzing individual traits but on the conjoint of them. LAI, FAPAR, and FVC showed similar patterns to stress since their highly physical correlation (Hu et al., 2020). The different patterns of Cw and Cab point to different drought and salinity resistance strategy components associated with these traits: LAI (and FAPAR/FVC) reflect the decrease in biomass due to stress, partly because stress directly and negatively impacts growth and partly because having a lower biomass decreases the evapotranspiration demands of the crop, which increases the resilience of the crop to deal with drought. Cw represents another pathway to reduce evapotranspiration demands, i.e. by reducing the amount of water per gram of leaves. Also this response may be a direct effect of the more negative pressure heads due to drought or due to increased osmotic pressures (due to salinity). It may also be part of the adaptive strategy of the crop to increase its

resilience. Cab also responds to drought and salinity, but in its own way, i.e. by adapting its photosynthetic capacity while being affected by a lower stomatal conductance (due to drought and/or salinity). See e.g. Wright et al. (2003) for a framework explaining these nitrogen-water interactions. In our revised manuscript, we will expand our discussion section to better explain these different responses of the five traits.

**References**

Hu, Q., Yang, J., Xu, B., Huang, J., Memon, M. S., Yin, G., Zeng, Y., Zhao, J., and Liu, K.: Evaluation of global decametric-resolution lai, fapar and fvc estimates derived from sentinel-2 imagery, Remote Sens., 12, https://doi.org/10.3390/rs12060912, 2020.

McKee, T. B., Doesken, N. J., and Kleist, J.: The relationship of drought frequency and duration to time scales, Proceedings of the 8th Conference on Applied Climatology, 1993, 179-183,

Tao, H., Borth, H., Fraedrich, K., Su, B., and Zhu, X.: Drought and wetness variability in the tarim river basin and connection to large-scale atmospheric circulation, Int. J. Climatol., 34, 2678-2684, https://doi.org/10.1002/joc.3867, 2014.

Wen, W., Timmermans, J., Chen, Q., and van Bodegom, P. M.: A review of remote sensing challenges for food security with respect to salinity and drought threats, Remote Sens., 13, https://doi.org/10.3390/rs13010006, 2020.

Wright, I. J., Reich, P. B., and Westoby, M.: Least-cost input mixtures of water and nitrogen for photosynthesis, Am. Nat., 161, 98-111, https://doi.org/10.1086/344920, 2003.

---

## Author Response (AR1)

**Dear Editor,**

We would like to submit our revised manuscript entitled "***Monitoring the combined effects of drought and salinity stress on crops using remote sensing in the Netherlands***" *(HESS-2022-50)*. The revision was based on two reviewers' comments. Our itemised responses are attached below and the changes made are also annotated in the revised manuscript. In order to facilitate the review, the reply is displayed in blue font, and the comment of reviewer is displayed in black. All revision was marked with the "Track Changes" function in Microsoft Word.

We hope that the revision is acceptable for publication. We deeply appreciate your consideration of our manuscript and look forward to your response.

Yours sincerely,
Wen Wen
PhD candidate
Institute of Environmental Sciences (CML), Leiden University
Room B2.10, Einsteinweg 2, 2333 CC Leiden, The Netherlands
Telephone: +31-071-5272727
E-mail: w.wen@cml.leidenuniv.nl

\* \* \* \* \* \* \* \* \* \* \* \* \* \* \* \* \* \* \* \* \* \* \* \* \* \* \* \* \* \* \* \* \* \* \* \* \* \* \* \* \* \* \* \* \* \* \* \* \* \* \* \* \* \*

**Response to Reviewer 1**

**General comment**

The combined effect of drought and salinity on crops is very important for food security under global change background. Remote sensing shows its advantage for large-scale applications. This paper used the sentinel-2 satellite data to conduct an analysis in this regard. The findings are interesting.

**Response/Action:** Thank you for your positive comments on our work. We appreciate that our work provides insights into monitoring drought and salinity impacts on crops by remote sensing on a large scale. Our itemized responses are provided below. In order to facilitate the review, the comment of the reviewer is displayed in black, and the reply is displayed in blue font.

**Major concerns**

Q1. Irrigation can mitigate the drought effect to a large extent. I would like to know how irrigation has influenced the analysis. There is no information reported in this regard.

**Response/Action:** We have added a remark to explain it to the discussion section of the revised manuscript. (Lines 241-247)

**Lines 241-247:** Although irrigation may modify the severity of drought impacts on crops, we have evidence that irrigation did not play a major role in the patterns found in this case since all croplands included within our research area have been identified as rainfed cropland according to the ESA/CCI land cover map in 2018 (https://maps.elie.ucl.ac.be/CCI/viewer/). In addition, while -in the area- farmers are known to irrigate their cropland, the Dutch government

announced a temporary national irrigation ban in various areas including our research area in 2018 (Perry de Louw, 2020) to spare water. Therefore, we assumed that irrigation management was absent during our study period.

Q2. Five different indicators were used to depict the health condition of different crops. I am wondering how the stress factors influence the final yield. Is it possible to have some discussion in this regard.

**Response/Action:** In our revised manuscript, we have highlighted the relationship between traits and the final yield in the discussion section to provide a wider application of our findings. (Lines 337-345)

**Lines 337-345:** In addition, our approach gives the insight to analyze the effect of stresses on yield based on the five traits, even though yield cannot directly be derived from remote sensing. Traits including Cab, LAI, and FAPAR, have been used (either separately or in combination) as a proxy for final yield estimates from remote sensing in many studies. For instance, NDVI - which is based on the combination of LAI and Cab- is extensively used to estimate crop yield (Huang et al., 2014; Mkhabela et al., 2011; Vannoppen et al., 2020). Also, LAI itself has been used for predicting the final yield (Sun et al., 2017; Dente et al., 2008; Doraiswamy et al., 2005). Meanwhile, Cab and FAPAR were also proven to be highly correlated with crop yield (López-Lozano et al., 2015; Ghimire et al., 2015). Thus, while yield cannot be estimated directly from remote sensing or ground truth data at the desired high spatial resolution, our indicators do relate to yield and can be used in more application-based contexts to inform on yield impacts.

**Minor comments**

Q3. Line 33, more deeply challenged.

**Response/Action:** We have revised "deeper challenged" to "more deeply challenged". (Lines 31-34)

**Lines 31-34:** With drought and salinity forecasted to increase spatially and in severity (Schwalm et al., 2017; Trenberth et al., 2013; Rozema and Flowers, 2008), and with predictions of higher co-occurrence around the world (Wang et al., 2013; Corwin, 2020; Jones and van Vliet, 2018), food production will be more deeply challenged by both stresses.

Q4. Line 37, delete 'of' and 'more than'

**Response/Action:** We have deleted 'of' and 'more than' from this sentence. (Lines 36-38)

**Lines 36-38:** Co-occurrence of drought and salinity stress is found to decrease the yield of spinach (Ors and Suarez, 2017) and the forage grass *Panicum antidotale* (Hussain et al., 2020) compared with the occurrence of one of these stresses only.

Q5. Lines 83-90, why was SPEI selected as the drought indicator rather than the others? What is the RD_new projection? Where are the precipitation and PET data from?

**Response/Action:** There are several common drought indices including the Palmer Drought Severity Index (PDSI), the Standardized Precipitation Index (SPI), and the Standardised Precipitation-Evapotranspiration Index (SPEI), etc., to evaluate drought events. PDSI, which is based on the water balance equation, has disadvantages due to autoregressive characteristics and its fixed temporal scale (Guttman, 1998). SPI, which is calculated from precipitation data,

shows better performance than PDSI on droughts detection thanks to its multi-scalar features (Hayes et al., 1999). Nevertheless, compared with these two common drought indices, the SPEI is a multiscale drought index based on precipitation and temperature data, and in this way, it has the advantage of detecting, monitoring, and assessing drought in multiple systems (Vicente-Serrano et al., 2010).

We have explained RD_new projection in the revised manuscript. (Lines 87-89)

**Lines 87-89:** The RD_new projection (EPSG:28992) is a projected coordinate reference system of the Netherlands. All maps were projected to RD_new projection to create consistent data layers.

The drought map was created by our group, and published in another study in the journal Science of the Total Environment (Chen et al., 2022). The precipitation data were fused based on remote sensing data and ground observations. The PET data was obtained from MODIS in 8-day composite dataset. More details can be found in Chen et al. (2022). (Lines 82-85)

Q6. Lines 124-129, Include some information about Sentinel-2 in the data description although it was pointed out in Fig. 1.

**Response/Action:** More information about Sentinel-2 has been added to the revised manuscript. (Lines 126-132)

**Lines 126-132:** The Sentinel-2 mission consists of two satellites equipped with the high-resolution Multispectral Instrument (MSI) in the same orbit. This sensor acquires 13 spectral bands (with varying spatial resolutions) in the visible and near-infrared spectrum at 5 days of revisit times (ESA, 2015). In our study, we used both the 10m and 20m Level 2A observations, downloaded from The Copernicus Open Access Hub (https://scihub.copernicus.eu/), to facilitate the requirement of the Sentinel Application Platform (SNAP) toolbox for both optical and near-infrared observations to be available for determining the functional traits. To create consistency across the bands, those with a 20m resolution (B5, B6, B7, B8A, B11, and B12) were resampled to the 10m resolution of B3 and B4.

Q7. Line 145, why was the biomass effect removed? Is this contradictory to the Cab*LAI and Cw*LAI at Line 142?

**Response/Action:** We have revised the text to make it more clear for readers. (Lines 149-154)

**Lines 149-154:** The biophysical processor within the SNAP toolbox derives the five traits, namely LAI, FAPAR, FVC, canopy chlorophyll content (CCC), and canopy water content (CWC), for each pixel from the Sentinel-2 top of canopy reflectance data. This processor utilizes an artificial neural network (ANN) approach, trained using the PROSAIL simulated database (Weiss and Baret, 2016). This training utilized canopy traits rather than leaf traits (estimated by multiplication with LAI) to improve their neural network performance. To obtain their leaf counterparts (Cw and Cab), to create fully independent variables, CCC and CWC thus need to be divided by LAI to obtain Cab (=CCC / LAI) and Cw (=CWC / LAI).

Q8. Line 151, What do you mean by 'due to the unbalance in the occurrence of stress conditions'?

**Response/Action:** We have revised it in the revised manuscript. (Lines 160-161)

**Lines 160-161:** Since the pixel counts of the six classes of stress combinations namely no stress, MD, SD, salinity, MD+MS, and MD+SS were (highly) different, drought and salinity were not considered as two independent factors.

Q9. Lines 163-173, More explanations are needed to illustrate the connotations of different indicators in the ANOVA analysis, to increase the readability. Probably this can be supplemented in the methodology section.

**Response/Action:** We have added more information on the different indicators in the notes of Table 1. (Lines 185-187)

**Lines 185-187:** Note: $F$ indicates the test statistic of the F-test; $p$ indicates whether the effect is statistically significant in comparison to the significance level ($p < 0.05$); Partial Eta Squared indicates the effect size of different factors; $R^2$ indicates the percentage that the model coincides with the data.

Q10. Line 200, Add some information for the different letters indicating the significance level.

**Response/Action:** All the significance levels are < 0.05 in Fig. 3 and Fig. 4. The letters in Fig. 3 and Fig. 4 indicate whether there is a significant difference among different stress groups based on the pairwise comparison. If the letter in one group is different from the other group, then a significant difference exists between these two groups. We'd like to clarify that it is a common way to show significant differences in bar plots throughout scientific literature as we have explained with the sentence 'Different letters in each panel indicate significant differences ($p < 0.05$)' in the caption.

Q11. Line 220 and Line 244, It was concluded at Line 220 that there is no additive effect for drought and salinity. Is it in contradiction to the severe effect of the co-occurrence of drought and salinity?

**Response/Action:** In fact, these two points are not contradicting. The effects of two factors can be either additive or interactive. If two factors are additive, then the effect of both factors (in this case drought and salinity) equals the sum of the effects of the individual factors. Thus, the reduction in a trait value by the combination of drought and salinity would equal the reduction due to drought plus the reduction due to salinity. If the effects of two factors are interactive, then the combined effect of two co-occurring factors does not equal the sum of the individual effects. That was the case here in which we found that particularly the MD+SS conditions led to major impacts on our functional traits.

We revised our statements to make this distinction clear. (Lines 262-263)

**Lines 262-263:** Specifically, we show that in real farmers' conditions, the co-occurrence of drought and salinity indeed can constitute a severe threat due to its interactive effects on crop growth.

Q12. Line 257, why was the drought effect mitigated? Please add more explanations.

**Response/Action:** More explanations have been added to the revised manuscript. (Lines 275-280)

**Lines 275-280:** Water stress impacts on photosynthesis and biomass of plants were extenuated by salinity since salinity enhances the synthesis of ATP and NADPH by promoting

photosynthetic pigments and photosystem II efficiency. The impacts of combined drought and salinity stress on plant growth, chlorophyll content, water use efficiency, and photosynthesis were less severe compared to drought alone. This indicates compensating effects on carbon assimilation due to osmotic adjustments induced by $Na^+$ and $Cl^-$ (Hussain et al., 2020). Thus, the detrimental effect of single drought stress on crop growth is considered to be mitigated by salinity.

Q13. Line 278, 'Considering the additional', remove the comma. Change 'promising' to 'probable'.

**Response/Action:** We have revised this sentence based on your suggestions. (Lines 300-301)

**Lines 300-301:** Considering the additional co-varying factors within our 'real-life' study, it is very probable that we were able to detect similar effects.

Q14. Line 283, 'In addition to facilitating the evaluation…'

**Response/Action:** We have revised 'being able to evaluate' to 'facilitating the evaluation'. (Lines 305-307)

**Lines 305-307**: In addition to facilitating the evaluation of crop performance during multiple stages of the growing season (in contrast to most destructive methods), remote sensing also allows a multi-trait approach to better understand the mechanisms involved in crop responses.

Q15. Line 285, distinctively

**Response/Action:** We have revised this sentence to make it easier to understand. (Lines 326)

**Lines 326:** Cab and Cw responses to drought and salinity were distinct from responses of LAI, FAPAR, and FVC (Fig. 3 and Fig. 4).

Q16. Line 288, understand

**Response/Action:** We have revised this sentence. (Lines 309-311)

**Lines 309-311:** Hence, given that individual crop traits may respond differently to drought and salinity reflecting its stress resistance and tolerance strategy, the evaluation of these distinct responses may help to understand this strategy.

Q17. Line 289, as compared to

**Response/Action:** We have revised this sentence according to your suggestions. (Lines 312-313)

**Lines 312-313:** In this study, Cw was consistently lower in all drought and salinity treatments as compared to no stress conditions in May, June, and July.

Q18. Line 291, In this respect

**Response/Action:** We have revised 'that' to 'this' in the sentence. (Lines 313-314)

**Lines 313-314:** In this respect, it is interesting that no decrease in Cw was observed at the end of the growing season, in September.

Q19. Line 292, the transpiration demand normally refers to the atmospheric demand, like VPD and incoming radiation. What do you mean here?

**Response/Action:** Here transpiration means the loss of leaf water vapor.

Q20. Please check the English writing more carefully to enhance the readability.

**Response/Action:** We have carefully revised the manuscript in terms of English writing to improve the readability.


[revised manuscript text omitted]

**Response to Reviewer 2**

**General comment**

Drought and salinity are considered to be the two main factors limiting crop productivity. Remote sensing enables the assessment of the impacts of extremes on crops, but it is seldomly used in the study of compound effects of drought and salinity stress. The novelty of this study is to assess the impacts of drought, salinity, and their combination on crop traits using multiple remote sensing observations and explore their relationships with stress timings and drought levels. The manuscript makes a contribution to the assessment of compound extremes' impacts using remote sensing and the writing is well organized. I suggest this manuscript should be accepted by HESS after minor revisions.

**Response:** Thank you for your positive comments about our work. We appreciate your support of the novelty and potential applications of compound extremes' impacts on a large scale. Our itemized responses are attached below. In order to facilitate the review, the comment of the reviewer is displayed in black, and the reply is displayed in blue font.

**Specific comments:**

Q1. In this study, only the 2018 case over the Netherlands was analyzed, are the conclusions robust? As the available data range from 2004 to 2018, are there any else cases to verify the conclusions? If there is no more cases, it is better to add the name of this case in the title.

**Response/Action:** From a detailed and thorough review (Wen et al., 2020), we found that at present nobody has investigated the combined effects of drought and salinity at such a scale using remote sensing. So, while this is an important novelty of our study, it also hampers verifying our conclusions with other cases. We use the years 2004 to 2018 to create a baseline of local hydrological conditions and to be able to evaluate the local deviations in those conditions for 2018 (given that 2018 was by far the year with the most extreme drought conditions in that period), but we cannot use the other years to verify our patterns as this would lead to circular arguments. At the same time, we believe our conclusions are robust in light of the approach we used and its consistency with results from small-scale local studies. We are currently validating our analysis for a much larger area in the USA, which should provide extra support for this study. The findings however are outside the scope of this particular paper.

Concerning the current limits of this case, we have revised the title in the revised manuscript. (Lines 1-2)

**Lines 1-2:** Monitoring the combined effects of drought and salinity stress on crops using remote sensing in the Netherlands.

Q2. Add a map of the crop distribution over the study region.

**Response/Action:** We have added a crop map of North-Holland province to Figure 2 (Fig.2c). (Lines 113)

**Lines 113:**

[Figure]

**Figure 2.** Map of the Netherlands overlaying a) drought and b) salinity in the Netherlands to show c) the co-occurrence of drought and salinity. The selected study area is indicated by black lines in panel c. d) The associated crop map of the study area.

Q3. Line 89-90: The standard deviation of SPEI is 1 (Vicente-Serrano et al 2010), why do you define drought when SPEI is less than -321?

**Response/Action:** We have explained the thresholds of drought classifications in detail in the revised manuscript. (Lines 89-92)

**Lines 89-92:** We defined -1 and -1.5 as daily thresholds for different drought severity classes according to previous classifications (McKee et al., 1993; Tao et al., 2014). Thus, (cumulative) SPEI for no drought should be between -214 to 0, SPEI for moderate drought should be between -321 to -214 and for severe drought, SPEI should be lower than -321 when calculated for the whole growing period (Fig. 2a).

Q4. The captions of Table 1, figure 3, and figure 4 should be described in detail, e.g. what are MD, MS, SS, ab, and abc short for in fig. 3?

**Response/Action:** We have added the meanings of all stress conditions to the caption of Fig.3 and Fig. 4 of the revised manuscript. All the different letters (e.g. ab) refer to the post hoc result from pairwise comparisons. It is a common way to show significant differences in bar plots throughout scientific literature as we have explained with the sentence 'Different letters in each panel indicate significant differences ($p < 0.05$)' in the caption. (Lines 211-213, 235-237)

**Lines 212-214:** Figure 3. Expressions of LAI, FAPAR, and FVC under various stress conditions in May, June, July, and September. Different letters in each panel indicate significant differences ($p < 0.05$). MD, moderate drought only; Salinity, salinity only; MD+MS, moderate drought and moderate salinity; MD+SS, moderate drought and severe salinity (MD+SS); SD, severe drought only.

**Lines 236-238:** Figure 4. Expressions of Cab and Cw under various stress conditions in May, June, July, and September. Different letters in each panel indicate significant differences ($p < 0.05$). MD, moderate drought only; Salinity, salinity only; MD+MS, moderate drought and moderate salinity; MD+SS, moderate drought and severe salinity (MD+SS); SD, severe drought only.

Q5. As figures 3-4 show the values of crop traits from May to September, it is better to show their standardized anomalies compared with climatology, which enables the comparison between different timings.

**Response/Action:** Unfortunately, due to several limitations, we cannot provide a climatology. In this case, we calculated traits based on Sentinel-2 which is only active since 2015. As such, sentinel-2 does not have observations (yet) to investigate the long-term behavior of those traits, thereby limiting us to define a climatology for no-stress conditions. While Landsat and other comparable satellites exist that could be used for this, we chose not to use this information because of the differences in wavelengths and resolutions among different satellites. In addition, we chose not to adopt a non-drought year (like 2015 or 2016) as a control (no stress condition) in this research, because we could not quantify the natural variation (in respect to a long-term climatology) based on those specific years. Hence, we were restricted in our approach to make a comparison to a control treatment. This comparison to the control provides the baseline for no-stress conditions that can be directly compared to the responses in stressed conditions. This avoids the need for including climatology.

Q6. What is the best explanation of the different responses among the five crop traits to such stresses?

**Response/Action:** In our revised manuscript, we have expanded our discussion section to better explain these different responses of the five traits. (Lines 307-311, Lines 326-336)

**Lines 307-311:** Each of the five traits is associated with different functions of plants that might be individually impacted by the different stresses. Therefore, focusing on only one individual metric (as commonly done; see Wen et al. (2020) for a review) limits our capacity to gain full insight into drought and salinity responses. Hence, given that individual crop traits may respond

differently to drought and salinity reflecting its stress resistance and tolerance strategy, the evaluation of these distinct responses may help to understand this strategy.

**Lines 327-337:** Cab and Cw responses to drought and salinity were distinct from responses of LAI, FAPAR, and FVC (Fig. 3 and Fig. 4). LAI, FAPAR, and FVC showed similar patterns to stress due to their highly physical correlation (Hu et al., 2020). The different patterns of Cw and Cab point to different drought and salinity resistance strategy components associated with these traits: LAI (and FAPAR/FVC) reflect the decrease in biomass due to stress, partly because stress directly and negatively impacts growth and partly because having lower biomass decreases the evapotranspiration demands of the crop, which increases the resilience of the crop to deal with drought. Cw represents another pathway to reduce evapotranspiration demands, i.e. by reducing the amount of water per gram of leaves. Also, this response may be a direct effect of the more negative pressure heads due to drought or due to increased osmotic pressures (due to salinity). It may also be part of the adaptive strategy of the crop to increase its resilience. Cab also responds to drought and salinity, but in its own way, i.e. by adapting its photosynthetic capacity while being affected by a lower stomatal conductance (due to drought and/or salinity). See e.g. Wright et al. (2003) for a framework explaining these nitrogen-water interactions.

---

## Author Response (AR2)

**Dear Editor,**

We would like to submit our revised manuscript entitled "*Monitoring the combined effects of drought and salinity stress on crops using remote sensing in the Netherlands*" *(HESS-2022-50)*. This revision is based on the comments provided by the reviewers based on our earlier revised manuscript. Reviewer 1 was satisfied with our revisions and recommended acceptance. Reviewer 2 mostly provided comments for clarification. We appreciate these suggestions. Our itemised responses are attached below and the changes made have also been annotated in the revised manuscript. In order to facilitate the review, the reply is displayed in blue font, and the comment of the reviewer is displayed in black. All revisions have been marked with the "Track Changes" function in Microsoft Word.

We hope that this revised manuscript is acceptable for publication. We deeply appreciate your consideration of our manuscript and look forward to your response.

Yours sincerely,
Wen Wen
Ph.D. candidate
Institute of Environmental Sciences (CML), Leiden University
Room B2.10, Einsteinweg 2, 2333 CC Leiden, The Netherlands
Telephone: +31-071-5272727
E-mail: w.wen@cml.leidenuniv.nl

\* \* \* \* \* \* \* \* \* \* \* \* \* \* \* \* \* \* \* \* \* \* \* \* \* \* \* \* \* \* \* \* \* \* \* \* \* \* \* \* \* \* \* \* \* \* \* \* \* \* \* \* \* \* \* \* \* \*

**# Response to Reviewer 2**

**Main comment**

The revisions to the manuscript have mostly addressed my major comments. However, considering there is only 2018 case analyzed here, it is better to discuss the uncertainty of these conclusions and explain why other cases are not included in the study. What is more, comparisons this study with other researches about the 2018 case could also be used to verify your conclusions. I am still confused about the timescale of drought. Is it fixed from April 1st to October 30th ? Considering the drought may not persist over the whole growing period, it may not be suitable to use cumulative thresholds (-214, -312) to classify drought.

**Response/Action:** Thank you for your comments. We now discuss the limitations of this study and explain the reason for not considering case studies from other years in a new dedicated section 4.4. (Lines 345-366) We are not aware of other studies executed for the year 2018.

With respect to the time scale of the drought: The drought map was created based on SPEI with a 3-months sliding time scale. The onset of the 2018 drought was from March to May across the Netherlands according to the result of our team (Chen et al., 2022). Crops are usually planted between April to May, and harvested before the end of October in the Netherlands. So, to be consistent with the crop growing season, we evaluated the drought impacts from April to October. Exactly because the drought may be intermittent over the growing period, the cumulative SPEI can capture the overall impacts of these droughts. The cumulative nature of

the metric also helps to evaluate impacts across multiple crops across large areas as different crops may respond more quickly or slowly to drought events. A cumulative SPEI captures all these responses. Moreover, the 2018 drought lasted up to the end of the growing season (October) as indicated by several other studies (Brakkee et al., 2022; Peters et al., 2020). In combination, we believe that using the cumulative SPEI gives reasonable insight to classifying drought and its impacts on crops.

**Lines 345-366:** 'The number of studies that evaluate the effects of drought and salinity stress on crops is limited (Wen et al., 2020). In general, studies focus on small-scale experimental studies under strictly control of all variables with only a limited number of crops (Hussain et al., 2020; Ors and Suarez, 2017). To our knowledge, this is the first study that uses satellite remote sensing to investigate drought and salinity impacts for a large area under real-life conditions necessary for constructing stress management policies.

In such real-life conditions, as investigated here, irrigation of crops is commonly applied as management practice during drought events to reduce the severity of drought impacts (Deb et al., 2022; Lu et al., 2020). In this study, however, we have evidence that irrigation did not play a major role in the patterns found since all croplands included in our research area were identified as rainfed cropland (according to the ESA/CCI land cover map in 2018; https://maps.elie.ucl.ac.be/CCI/viewer/). In addition, while farmers in the area are known to irrigate their cropland, the Dutch government announced a temporary national irrigation ban in 2018 (for various areas including our research area) to spare water (Perry de Louw, 2020). As a consequence, we could not analyze the impacts of irrigation management on the combined effects of drought and salinity. This might potentially be solved by investigating other drought historic events with moderate severity in Europe, such as the year of 2003 (Ciais et al., 2005) or 2015 (Ionita et al., 2017) in Europe, when such a ban was not executed. Unfortunately, satellite remote sensing observations with the required 20-30m resolutions of these events are limited, as Sentinel-2 was only launched in 2015 and the Landsat satellites provide a too coarse temporal resolution.

Likewise, impacts of salinity and drought are moderated by crop selection. Traditionally, farmers do not plant highly vulnerable crops in moderate/high salinity areas. In fact, we found crops sensitive to salinity such as apple (Ivanov, 1970) and broccoli (Bernstein and Ayers, 1949) to be abundant in non-saline areas but only little in saline areas. To ensure an accurate evaluation of salinity impacts, we only investigated those crops with a significant abundance in all available stress conditions. More sensitive crops might even respond more strongly.'

**Specific comments:**

Q1. L29: drought and salinity "stress".

**Response/Action:** We have revised this sentence according to your suggestion. (Lines 29-31)

**Lines 29-31:** 'Of these stresses, drought and salinity stress have been identified as the two main factors to limit crop growth, affecting respectively 40% and 11% of the global irrigated areas (Dunn et al., 2020; FAO, 2020).'

Q2. L33 Rank the references.

**Response/Action:** The references have been reranked. We have revised the citations sort order over the whole manuscript. (Lines 31-34)

**Lines 31-34:** 'With drought and salinity forecasted to increase spatially and in severity (Rozema and Flowers, 2008; Schwalm et al., 2017; Trenberth et al., 2013), and with predictions of higher co-occurrence around the world (Corwin, 2020; Jones and van Vliet, 2018; Wang et al., 2013), food production will be more deeply challenged by both stresses.'

Q3. L36: Co-occurrence of drought and salinity stress is found to decrease… "more" compared the individual stress only.

**Response/Action:** We have revised this sentence according to your suggestion. (Lines 36-38)

**Lines 36-38:** 'Co-occurrence of drought and salinity stress is found to decrease the yield of spinach (Ors and Suarez, 2017) and the forage grass *Panicum antidotale* (Hussain et al., 2020) more compared with the occurrence of one of these stresses only.'

Q4. L55-57: Please conclude the specific studies about the impact of drought/salinity stress on crops. It is better to put this study to the background of the research about assessing the impacts on crops using RS techniques.

**Response/Action:** We have added specific studies in background information. (Lines 55-60)

**Lines 55-60:** 'Canopy chlorophyll content and mean leaf equivalent water thickness (EWT) of maize differed remarkably under drought stress using hyperspectral remote sensing data (Zhang and Zhou, 2015). Using a look-up-table approach, LAI and chlorophyll content of wheat obtained from a radiative transfer model showed potential to assess drought levels (Richter et al., 2008). However, while there have been several attempts to monitor the response of crop health with either a drought or salinity focus, not much research has taken these factors into account simultaneously (Wen et al., 2020).'

Q5. L114-115 & L211-213: Add "in 2018" to the figures' captions.

**Response/Action:** We have revised the captions of the figures according to your suggestion. (Lines 117-118, Line 215, Line 239)

**Lines 117-118:** 'Figure 2. Map of the Netherlands overlaying a) drought and b) salinity to show c) the co-occurrence of drought and salinity in 2018. The selected study area is indicated by black lines in panel c. d) The associated crop map of the study area in 2018.'

**Line 215:** 'Figure 3. Expressions of LAI, FAPAR, and FVC under various stress conditions in May, June, July, and September 2018.'

**Line 239:** 'Figure 4. Expressions of Cab and Cw under various stress conditions in May, June, July, and September 2018.'

Q6. L138-147: What are the spatial resolution and time scale of LAI, FAPAR, and etc?

**Response/Action:** All traits are at 10 m resolution and each monthly trait estimate was derived by the biophysical processor within SNAP. We have clarified the related information in section 2.3. (Lines 152-154)

**Lines 152-154:** 'The biophysical processor within the SNAP toolbox derives the five traits, namely LAI, FAPAR, FVC, canopy chlorophyll content (CCC), and canopy water content (CWC), for each pixel from the Sentinel-2 top of canopy reflectance data at a 10m-resolution for each month.'

Q7. L163: What is "hoc test"?

**Response/Action:** The post-hoc test is an integral part of an ANOVA. While the ANOVA identifies that significant differences occur among at least two of the groups, it does not indicate which groups differ significantly. A post-hoc test evaluates which groups differ significantly from each other within the ANOVA.

Q8. L164: As you mentioned "ANOVA" many times, explain the method more in details.

**Response/Action:** ANOVA refers to the analysis of variance. A two-way ANOVA is a test to compare the difference between groups based on two factors. In this study, we adopted a two-way ANOVA to evaluate the main effects of the factors 'stress' and 'date' as well as the interaction of stress and date. We have explained ANOVA in section 2.4. (Lines 164-166)

**Lines 164-166:** 'Instead, two-way analysis of variance (ANOVAs) was applied to test the main effects and the interactive effect between stress combinations (consisting of 6 levels) and time (5 months) on each individual crop trait.'

Q9. L347-348: Add "during 2018 over the Netherlands".

**Response/Action:** We have revised this sentence according to your suggestion. (Lines 368-369)

[revised manuscript text omitted]